# Repeated measures of decaying wood reveal the success and influence of fungal wood endophytes

Yanmei Zhang,[1] Zhuobing Peng,[2] Zewei Song,[3,4] Jonathan S. Schilling[1]

**ABSTRACT** Predicting wood decomposition is challenging due to complex successional dynamics among decomposers that colonize and defend wood territory. This starts with saprotrophic fungi that reside latently in healthy wood until trees senesce, but these "endophytes" are rarely considered an endogenous wood trait that might improve predictions for decomposition rates or fates. Here, we used repeated measures to track the decomposition of paper birch (*Betula papyrifera*) and red pine (*Pinus resinosa*), assessing wood properties and microbial succession over 5 years in a northern forest (Minnesota, USA). We compared fungi and bacteria present in sound wood (endophytes) versus those arriving as external colonizers, and we used relevant treatments to vary accessibility for colonizers (ground contact versus aboveground; bark off versus bark on). Over 5 years, accessibility treatments had a significant effect on decay rates and fungal community succession. Wood rot type was unanimously white rot (lignin-degrading fungi), but fungal dominance was treatment-specific. Most dominant fungi could be traced to operational taxonomic units (OTUs) present as endophytes in sound wood, suggesting that treatments affected endophyte competition more than external colonizer success, even in ground contact. Although fungal communities lost diversity (Shannon index) as certain taxa became dominant, bacterial communities converged irrespective of treatment, without notable co-occurrence with fungi and without losing diversity, suggesting a decoupled dynamic. The results imply a strategic benefit for saprotrophic fungi to colonize trees as endophytes, and they support including fungal endophytes along with predictors of their competitive success as "plant" traits to improve predictive models.

**IMPORTANCE** Establishing this persistence and influence for endophytic saprotrophs has not been possible without repeated measures in a long-term study. We believe our findings are significant for two key reasons. First, they link community succession to a "trait" in wood that may be more predictable—governed by the living tree as the "gate-keeper"—compared with predicting assembly history for external colonizers. Second, they highlight a new avenue toward developing a predictable trait for wood decomposition that could improve Earth Systems modeling, which has historically been challenged in predicting carbon sequestered/released by wood, where most of Earth's aboveground biotic carbon resides.

**KEYWORDS** wood decomposition, sequencing, decomposer, functional ecology, endophyte, colonization, inoculum potential, priority effect

**Peer Reviewer** Yuanyuan Bao, Nanjing Forestry University, Nanjing, China

Address correspondence to Jonathan S. Schilling, schillin@umn.edu.

The authors declare no conflict of interest.

See the funding table on p. 15.

Deadwood contains approximately 8% (or 73 Pg) of global forest carbon (1). It is estimated that about 10.9 (± 3.2) Pg of carbon is released annually from deadwood decomposition (2). This carbon, derived from lignocellulose, is primarily liberated into the atmosphere as billions of tons of $CO_2$ (3), comparable with the carbon emissions from fossil fuel combustion (4). The remaining carbon during biological wood decomposition

is either taken up by fungal biomass or solubilized into soils as organic carbon and other companion elements (5). Therefore, deadwood decomposition plays a crucial role in forest nutrient cycling (5), soil fertility maintenance (6), and the preservation of soil microbial diversity (7–9). It is paramount to understand the factors that control the turnover rate or fate of deadwood decomposition (10) and to incorporate this understanding into forest ecosystem predictive models (11).

Deadwood decomposition is primarily governed by microbial processes that introduce more local-scale variation than is explained by broad-scale variables such as climate or substrate quality (12, 13). Fungal trait-based models that consider microbial life-history strategies and their environmental responses during wood decomposition are expected to improve broad-scale predictions (14). Fungi that dominate wood deconstruction not only have different growth and decay rates (14) but also have distinct nutritional strategies that have non-redundant consequences on carbon release —some avoid lignin (brown rot; soft rot), and others mineralize lignin to access and metabolize carbohydrates (white rot) (15–17). In the field, fungal community assembly history or the sequence and timing of species arrival seem to be contingent on both abiotic and biotic conditions. Given that wood-decomposing fungi are thought to have lower host-specificity than other saprotrophs (18), these historical contingencies in wood decomposer assembly can lead plant trait-based predictions in unexpected directions regarding community succession (19, 20), thus creating discrepancies between observed and predicted wood decay rates in models (13, 21). This range of lignin selectivity among dominant fungi thus adds uncertainty in predicting the fate of woody lignin, where nearly one-third of Earth's aboveground biotic carbon is bound (22).

The successional dynamics of wood decomposers are fairly well-known but have largely been stitched together as time sequences using chronosequence or other snapshot approaches, where it is difficult to track the persistence of taxa over time. Succession in deadwood begins with early microbial colonizers that capture and defend territory in a three-dimensional wood space that has a lower surface area-to-volume than leaves or needles. Early (priority) colonizers can dominate wood as territory and alter success for later arriving species either by direct competition/obstruction or by indirect promotion/inhibition (13, 23–25). These priority effects have been tested via inoculation (26, 27), but they may initiate with endophytes, organisms already present in healthy wood without causing symptoms. Fungal endophytes, in particular, include stress-tolerant latent saprotrophs that become liberated as trees weaken and senesce (28–31). The relative abundance of DNA contributed by endophytes, defined here using the convention of being present without evident negative effects in living plant tissues, is low in sound wood, but the diversity of saprotrophic taxa is often surprisingly high (30). When conditions become favorable, saprophytic endophytes can gain and defend territory in deadwood, often creating pseudosclerotial plates (aka "spalting") to maintain boundaries (32).

A major uncertainty about wood endophytic fungi is their persistence and "downstream" influence on succession, a dynamic that varies in wood (33) and that would be best studied by using repeated measures and by varying access to external colonizers as deadwood transitions to the forest floor. Many endophytic-turned-saprotrophic fungi are stress-tolerant (S-selected) and poor combatants (34), and their persistence likely depends on their ability to fend off external colonizers, which may arrive from soil mycelial inoculum or from airborne spores. Contact with soil and the presence of bark are both naturally variable elements of deadwood that control moisture availability and also control accessibility for external colonizers. Ground contact influences log wood decay rates (33%–80% slower in aboveground (35), as an example), and position (standing vs. downed) has been shown to have a substantial effect on forest carbon release (11). Similarly, bark is known to influence wood decomposition rates (36) and microbial assembly histories (37).

Another uncertainty in community assembly in deadwood that would benefit from repeated measures is the role of bacteria. Bacteria are also present as endophytes and

participate in colonization dynamics, and although bacteria are often assumed to be a consequence of fungal-derived conditions, they could influence fungal colonizer success and thus influence wood decay (38–40). We know that fungi can decompose wood in the absence of bacteria, generally faster than they can decompose wood when inoculated into non-sterile wood (30), and as a protocol, standardized wood durability tests use sterile conditions with single-strain fungal inoculations (41). Bacteria can degrade specific components of wood (e.g., pit membranes) (42) but generally cannot degrade wood to completion, at least within a time frame of decades (41). This does not necessarily imply, however, a certain directionality in the fungal-bacterial relationship during wood decomposition. Fungi certainly may be responsible for shaping bacterial communities as a function of carbon sources released by fungi (43, 44), directly by fungal inhibition or obstruction (45, 46), or even by facilitation as "highways" (47). Bacteria, however, could also be shaping fungal success by fixing and supplying nitrogen (48, 49), antagonizing some fungi but not others (38), or limiting certain fungal strategies (e.g., brown rot ROS-based depolymerization) by "stealing" solubilized carbon before the fungus can consume it (38).

To address these uncertainties in endophyte influence, we used a repeated measures approach in a manipulative field trial to track succession over five years of wood decay. We annually sampled the same tree logs to track the assembly of fungal and bacterial communities, in parallel with their consequences on wood physiochemistry. This study, as with any fungal community study, cannot yet deploy metagenomics techniques using assemblies from shotgun sequencing data, given the large whole genome sizes for these dominant filamentous fungi, along with the lack of annotations. This inability to do metagenomics does not hinder testing our hypothesis with amplicon data, given that the focus is on the persistence of taxa, including endophytes. A repeated measures approach allowed us to describe succession *in situ*, rather than relying on simultaneously and randomly sampled logs along a bole decay class chronosequence (20, 50, 51). Initiating this sequence with sound wood enabled us to determine the endophyte effect strength and persistence within discrete logs, including angiosperm *B. papyrifera* and conifer *P. resinosa* species. We also used naturally relevant treatments to vary external colonizer accessibility and microclimate conditions: bark off vs. bark on; ground contact vs. aboveground. Our specific questions were (i) how temporally persistent fungal endophytes and subsequent colonizers were, as a function of relevant colonization barriers, and (ii) how these patterns tracked for bacteria alongside the fungi. We hypothesized that endophytes would exert influence and maintain persistence longer (in terms of decay stage, not time) in bark on logs and aboveground placement. We also hypothesized that bacterial communities would shift toward dominant taxa in parallel with dominant fungal taxa, assuming an active and direct correlation (irrespective of directionality) between fungal and bacterial presence.

## MATERIALS AND METHODS

### Five-year field study

This study was conducted in the Experimental Forest at the University of Minnesota Cloquet Forestry Center (46.70° N; −92.52° W; elevation 385 m) on a site dominated by paper birch (*B. papyrifera*), red pine (*P. resinosa*), and white spruce (*Picea glauca*). In October 2010, 10 healthy paper birch trees and 10 healthy red pine trees (7–9 cm diameter at the base, >2.5 m height; >10 years old) were felled. After removing the branches, each tree was cut into four 50 cm logs (log A, B, C, D), with a 5–7 cm short disc cut from the middle (Fig. 1). A total of 80 discs were obtained in 2010 and immediately frozen at −80℃ as time zero samples (sound wood).

For the wood decay treatments, we removed bark from the logs B and put them alongside the logs A in an east-west transect in ground contact, spaced at 1 m intervals and oriented with the cut ends facing north-south. The logs C were hung with the cut ends facing up-down, parallel to tree stems, at a distance of 1 m above the ground on

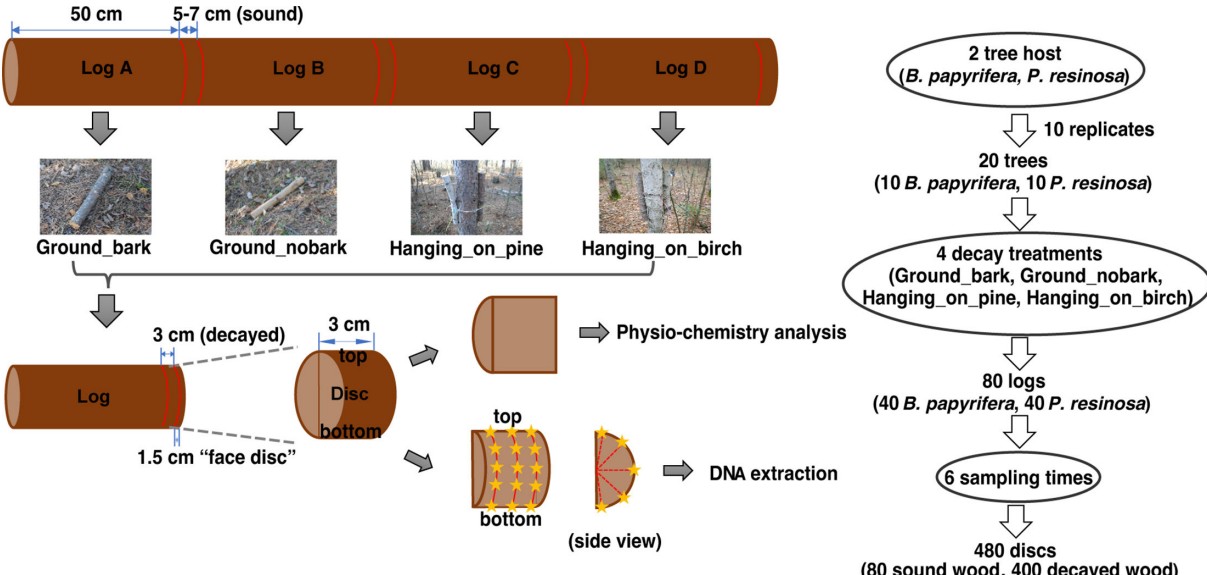

**FIG 1** Schematic representation of wood decomposition field design in Cloquet Forest Center. Ten tree replicates of *B. papyrifera* and *P. resinosa* were felled and set up for field decomposition, as described in Materials and Methods. About 5–7 cm of wood discs from 2010 were used as a time 0 control (sound wood). About 3 cm of wood discs were taken from one end of each log over the following 5 years. The factors used to study their effects on microbial community composition are listed on the right. The yellow stars indicate the drilling positions for wood sawdust for DNA extraction.

live birch trees, using nylon rope and propping on a galvanized nail driven into the tree. The logs D were hung on live pine trees, creating aboveground treatments on self- vs. non-self "host" trees (Fig. 1).

From 2011 to 2015, we sampled logs by removing a 1.5 cm "face disc" from the south- or ground-facing end, and then cutting a 3 cm (thickness) internal disc using an ethanol-sterilized handsaw. A total of 400 discs were collected over 5 years among the species and treatments. At each sampling time (annually; fall/autumn), discs were immediately cold-transported for 2 h and frozen at −80°C. When processing these frozen discs, they were surface-cleaned with 70% ethanol, and the bark was removed using a sterile scalpel. Frozen wood discs were then cut in half along the diameter that intersected the disc edge either at the original ground contact point or the "host" tree contact point, preparing one half for DNA analyses and the other half for wood physicochemical characterization (Fig. 1). To reduce the risk of contamination, we started by thoroughly cleaning the workbench and tools used for cutting wood, such as handsaw blades and chisels, with DNA AWAY surface decontaminant (Thermo Scientific, Waltham, MA, USA), followed by 70% ethanol between uses. The scalpels and 1/8-inch diameter drill bits were sterilized at 180°C for 4 h.

## Wood physiochemistry

To track the decay rate and assess the dominant rot type, we calculated the lignin loss to density loss ratio (L:D), as well as measuring carbohydrate profiles. Wood density was measured as dry mass (g; 48 h, 100°C) per fresh volume ($cm^{-3}$; water displacement), with density losses calculated relative to sound wood sampled at time zero. Bole decay classes for birch and pine were delineated using Appendix 2 of Harmon et al. (52). A Weibull residence model (53) was used to predict "years remaining" residence times for each tree log.

Oven-dried samples were Wiley-milled to 20 mesh for acid-insoluble lignin and structural carbohydrate profiling (glucan, xylan, arabinan, mannan, and galactan) (54). The L:D value was calculated to indicate the nutritional mode and lignin selectivity of dominant fungi (> 0.8 lignin-selective white rot fungi; < 0.8 carbohydrate-selective

brown rot and soft rot fungi) (16, 55). L:D comparisons were restricted to decay class II/III, where L:D values were more reliable to distinguish rot type (16).

## DNA extraction and sequencing

We drilled wood sawdust in each disc (Fig. 1) and extracted DNA from 400 to 500 mg of sawdust using the Qiagen PowerLyzer kit with slight modification (56). We quantified DNA using a Qubit 2.0 fluorometer with Qubit dsDNA HS Assay Kit (Life Technologies, Carlsbad, CA, USA). We amplified the V4 region of the 16S rRNA gene with 515F/806R primers and the ITS2 gene with 5.8 SR/ITS4 primers. The library preparation and amplicon sequencing were performed by BGI using DNBSEQ-G99 (BGI, Shenzhen, China) at PE300 mode. The raw sequencing data were processed using the Python-based Snakemake pipeline to generate an OTU table as outlined previously (57). Briefly, the sequencing adaptors and primers were first removed by cutadapt (58), and then, a 41 bp tail representing the large subunit (LSU) was removed from the end of the reverse primer. Sequencing reads with low quality (maximum expected error rate >1 and minimum length <100 bp) were discarded by Vsearch (59). The forward reads R1 and the reverse reads R2 were separately aligned to SILVA (v132) or UNITE (v7.2) database, with a 97% similarity in best mode using BURST (60). R1 and R2 alignment were compared to generate a consensus alignment, which was used to calculate the operational taxonomic unit (OTU) table with taxonomic assignment.

## Bioinformatics

For fungal ITS2 sequencing, DNA samples from a total of 463 wood disc samples were successfully sequenced, and 17 decayed wood samples were not able to be amplified, with no pattern among treatments or time points to explain amplification failures. The OTU table was first filtered to remove six non-fungal reads. Notably, sound wood had significantly fewer fungal reads than decayed samples, with an average of 311,920 versus 1,503,250 (Wilcoxon test, $P < 0.0001$). Subsequently, samples with low-quality sequencing reads (< 1,000 fungal reads) were removed. Additionally, low-abundance OTUs, represented by <10 reads in any given sample, were removed. After this filtering, the fungal data set had 460 samples with an average of 1,305,847 reads per sample (1,014–10,503,848). We assigned fungi to ecological categories at the genus level using the FungalTrait database (61). The bacterial data set had 478 samples that were successfully sequenced. The chloroplast and mitochondrial reads were first removed from the data set. Notably, these chloroplast and mitochondrial reads comprised about 80.45% of the total reads in sound wood, whereas only 1.23% in decayed wood. Consequently, after removing chloroplast and mitochondrial reads, 35 out of 80 sound wood samples (44%) had fewer than 1,000 reads, whereas only 2 of the 398 decayed samples (0.5%) had fewer than 1,000 reads. Notably, sound wood had significantly fewer bacterial reads than decayed samples, with an average of 60,776 versus 171,951 (Wilcoxon test, $P < 0.0001$). We only removed samples with <100 reads for quality filtration to retain most sound wood. Furthermore, low-abundance OTUs represented by <10 reads in any given sample were removed. After quality control and filtering, the bacterial data set consisted of 476 samples with an average of 153,617 reads per sample (109–1,885,330). The low fungal or bacterial reads in sound wood samples are attributed to the biological truth of low relative abundances of microbial DNA compared with host DNA in living trees/wood, not sequencing bias. To account for the read depth differences between sound and decayed wood samples, we (i) transformed data to relative abundance in units per mille by dividing each OTU in each sample by the total sequencing depth for the respective samples and then multiplying by 1,000 or (ii) performed rarefaction with 100 iterations for sound wood samples (fungi: 5,193; bacteria: 1,056) and decayed wood samples (fungi: 19,501; bacteria: 12,524), separately. The trends observed in fungal and bacterial diversity throughout the decay process and the direction of treatment effects were similar when comparing these two data normalization methods (Table S1). We opted for the relative

abundance transformation to avoid discarding additional information in the decayed samples with high reads.

## Statistical analyses

Statistics and data visualizations were conducted using R version 4.4.0. All reported values are presented as means ± standard errors. Unless otherwise noted, statistical significance was considered at $P < 0.05$.

The effect of tree host on density loss, mass residence time, and the chemical composition of sound wood was assessed by the Wilcoxon test. The repeated measures Friedman test was used to evaluate the effect of decay time or treatment on density loss or mass residence time, whereas samples were paired in other groups accordingly, followed by the Bonferroni-corrected pairwise paired Wilcoxon test. The effect of treatment on the L:D ratio was analyzed by the Kruskal test followed by a Wilcoxon test. The effect of treatment on carbohydrate loss was analyzed by a one-way repeated measures ANOVA test followed by a Bonferroni-corrected paired $t$-test. Spearman's correlation and general linear models were employed to test the relationships between carbohydrate loss and density loss.

The fungal or bacterial OTU table before removing the low-abundance OTUs (represented by <10 reads in any given sample) was used to calculate the alpha diversity indices, including richness and Shannon. The effect of decay time on diversity indices was tested using Kruskal-Wallis tests, followed by a pairwise Wilcoxon test. The effect of treatments (two group comparisons, such as bark on/bark off, ground contact/above-ground) on the diversity indices was tested using the Wilcoxon test.

Community dissimilarity between microbial communities was measured using Bray-Curtis distances in the "vegan" R package. The overall differences in the composition of microbial communities were visualized with principal coordinates analysis (PCoA) ordination plots. The effect of tree host, decay time, treatment, and their interaction was tested with permutational multivariate analyses of variance (PERMANOVA; adonis2). The effect of decay time, treatment, and interaction was also independently evaluated for each tree host using PERMANOVA. The differences in community dispersion were tested using dispersion analysis (betadisper) and ANOVA on resulting dispersion values. Kruskal-Wallis tests and Wilcoxon tests were used as appropriate to compare community distances between treatment groups. To characterize the distinctive fungal genus in each treatment compared with the ground contact, bark on samples (control), the differential analysis of the dominant genus (>7% relative abundance in one of the treatments over decay time) was performed by "edgeR" R package. Fungal endophytes were designated as the taxa shown at time zero in at least one of the 10 tree logs used in each treatment. The read count percentage of endophytes was calculated as endophyte persistence. The effect of decay time or treatment on endophyte persistence was assessed by the Friedman test, followed by the Bonferroni-corrected pairwise paired Wilcoxon test. The top 10 abundant fungi from each decay treatment at each sampling time were linked to their presence in the sound wood. To determine the relative effect of endophyte persistence and treatments, the distance between ten biological replicates and that between four treatments was compared.

The co-occurrence network analysis of fungi and bacteria was conducted at both the OTU and genus levels, separately. The OTU tables for fungi and bacteria were transformed to relative abundance separately. OTUs with low abundance, defined as appearing in fewer than five samples and having a relative abundance of less than 0.5%, were excluded for both fungi and bacteria. The resulting fungal and bacterial OTU tables were then merged on a per-sample basis. Next, pairwise Spearman correlations were calculated for the combined data, covering fungal–fungal, bacterial–bacterial, and fungal–bacterial associations. Only those correlations with estimates less than −0.8 or greater than 0.8, along with a Benjamini-Hochberg adjusted $P$-value of less than 0.01, were included in the network analysis. Finally, the correlation matrix served as the input for converting to an adjacency list using the "igraph" R package. The same approach was

applied at the genus level. The fungal-bacterial co-occurrence networks were visualized using Cytoscape (62).

## RESULTS

### Wood decay extent and rot types

Paper birch decayed significantly faster than red pine over 5 years of decomposition in the field ($P < 0.0001$), and decay extent varied depending on ground contact and bark presence (Fig. 2a and b). The presence of bark accelerated the decay of ground contact logs in birch ($P < 0.0001$) and pine ($P = 0.039$). In birch, ground contact accelerated decay relative to aboveground treatment logs hanging on pine as the "host" tree ($P = 0.002$), but not on birch as the host. Log residence times, in terms of predicted time to reach 10%, 25%, and 50% mass loss, varied accordingly by treatment for decaying birch logs (e.g., less time for birch with bark on) but not for decaying pine logs (Fig. S1).

Given the unequal decay rates between tree species, the L:D "comparables" (to compare "apples-to-apples" at the same decay stage, irrespective of time) were year 2 birch and year 5 pine (Fig. 2c). Although L:D in decay class II included some brown rot, notably for aboveground logs, the majority of the logs of both tree species had L:D higher than the 0.8 threshold, reflecting a consistent white rot dominance. The L:D averages within decay classes II/III were 1.216 (se 0.088) for year 2 birch and 1.099 (se 0.103) for year 5 pine, reflecting faster lignin removal (more lignin-selective), relative to carbohydrate removal. There was, however, no statistical difference among treatments for L:D values for either birch or pine.

The sound wood (non-decayed) of birch and pine differed in hemicellulose and lignin contents (Table S2). Typical of angiosperms versus gymnosperm conifers, birch had more xylan than pine, and pine had more mannan, galactan, arabinan, and lignin than birch ($P < 0.001$). Spearman's correlations showed that losses of glucan, xylan, and lignin were positively correlated with density loss for both birch and pine (Fig. S2), but only birch had significant treatment effects on glucan loss ($P = 0.018$), xylan loss ($P = 0.003$), and lignin loss ($P = 0.031$).

### Fungal community diversity

The ITS2 gene sequencing revealed 3,750 total fungal OTUs, with 1,353 OTUs in sound wood and 3,565 OTUs in decayed wood. The overlap of 1,168 OTUs between sound and decayed wood indicated a significant persistence of OTUs from sound wood to decayed wood. The additional OTUs in decayed wood represent an influx from external colonizers and resulted in a significant increase in species richness in all decayed wood. A significant decrease in the Shannon index, however, made it apparent that certain fungal taxa became relatively more abundant (dominant) in decaying wood (Fig. S3a and b). Specific to treatments, the richness and Shannon index were significantly higher in ground contact logs, particularly bark-off logs, compared with aboveground logs in decayed birch and pine (Fig. S3c and d), indicating colonization by soil-borne fungi and the role of bark as, we assume, a physical barrier. Consistent with more advanced decay stages of birch versus pine, Shannon indices were lower in birch than in pine, overall (Fig. S3b and d).

Birch and pine had statistically distinct fungal communities ($P = 0.001$, $R^2 = 0.079$) (Table S3), both in sound wood ($P = 0.001$, $R^2 = 0.058$) and, as an increasingly diverging pattern, in decayed wood (Fig. 3a and b; $P = 0.001$, $R^2 = 0.108$). Fungal communities in decayed wood differed significantly from those in sound wood (Fig. 3a and c; $P = 0.001$, $R^2 = 0.060$). The effect of decay time on fungal communities was more pronounced in the more heavily decayed birch than in pine (birch: $P = 0.001$, $R^2 = 0.119$, pine: $P = 0.001$, $R^2 = 0.064$). The treatments began to exert influence once decay started and became more pronounced as decay progressed (Fig. 3c and d; sound wood: $P = 0.067$, $R^2 = 0.035$; decayed wood: $P = 0.001$, $R^2 = 0.110$), reflecting a sustained effect of treatments on the development of fungal communities over time (Fig. S4).

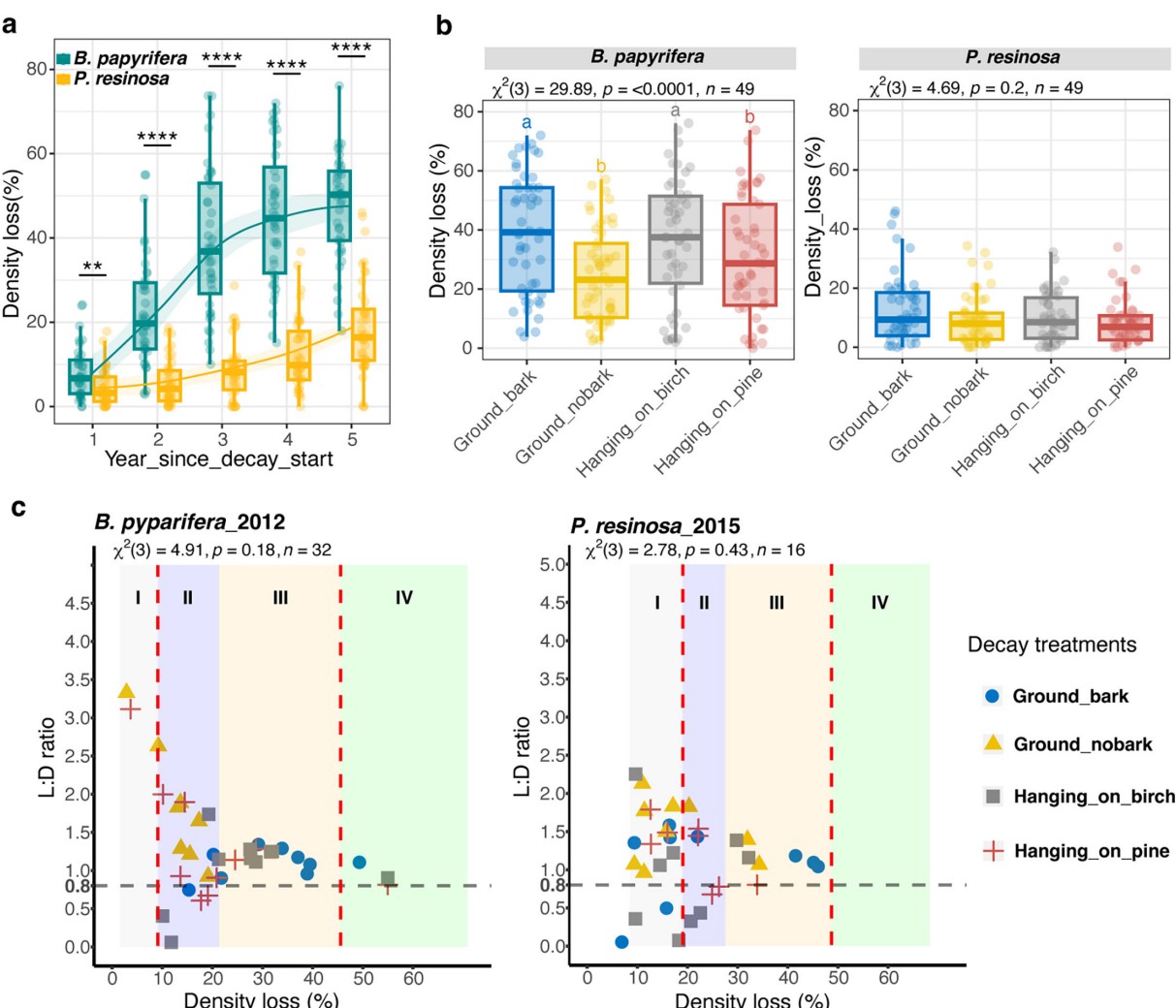

**FIG 2** Repeated measures of wood decomposition of *B. papyrifera* and *P. resinosa* in treatments with varied colonizer accessibility. (a) Density loss of *B. papyrifera* and *P. resinosa in situ* over 5 years. The stars indicate the tree host effect at each sampling year by the Wilcoxon test. *$P < 0.05$, **$P < 0.01$, ***$P < 0.001$, ****$P < 0.0001$. The x-axis represents the time since decay start (unit: Year). (b) Treatment effects on density loss in *B. papyrifera* and *P. resinosa*. Boxes represent 25th to 75th percentiles and display the median values (bold line in box). The *P* values are from the Friedman test for each tree host, and the lowercase letters are from the pairwise paired Wilcoxon test. (c) Ratio of lignin loss to density loss (L:D) as a function of density loss in *B. papyrifera* after 2 years of decay and *P. resinosa* after 5 years. L:D ratio was shown on a scale of the bole decay classes (I, II, III, IV, V), which was assigned based on density loss for each tree species. For *B. papyrifera*, decay class I (1.55%–9.17%), decay class II (9.18%–21.35%), decay class III (21.36%–45.63%), decay class IV (45.64%–70.83%), and decay class V (>=70.84%). For *P. resinosa*, decay class I (18.54%–19.02%), decay class II (19.03%–27.56%), decay class III (27.57%–48.78%), decay class IV (40.79%–68.29%), and decay class V (>=68.30%). The gray-dotted line indicated the threshold of 0.8 for discriminating rot type (> 0.8 lignin selective white rot fungi; < 0.8 carbohydrate selective brown rot and soft rot fungi). The red-dotted lines indicated the range of decay classes (between class II/III) where L:D values were more reliable to distinguish rot type and compared among treatments. The *P* values indicate the treatment effects from the Kruskal-Wallis test for each tree host.

## Fungal succession by taxa and by guild

Fungi in Ascomycota and Basidiomycota dominated in both birch and pine, with an initial rise of Basidiomycota fungi during early decay stages and a subsequent transition to Ascomycota fungal dominance at later decay stages (Fig. S5a). Fungi in Mucoromycota and Mortierellomycota present at relatively lower abundances in sound wood were generally lost within the first decay year.

At the order level (Fig. 4a), the composition of fungal communities in sound wood was more even (higher Shannon index) than in decayed wood, where dominant orders were apparent and more treatment-dependent. At the genus level, fewer assignments could be made relative to order assignments, but the same pattern of lost diversity and

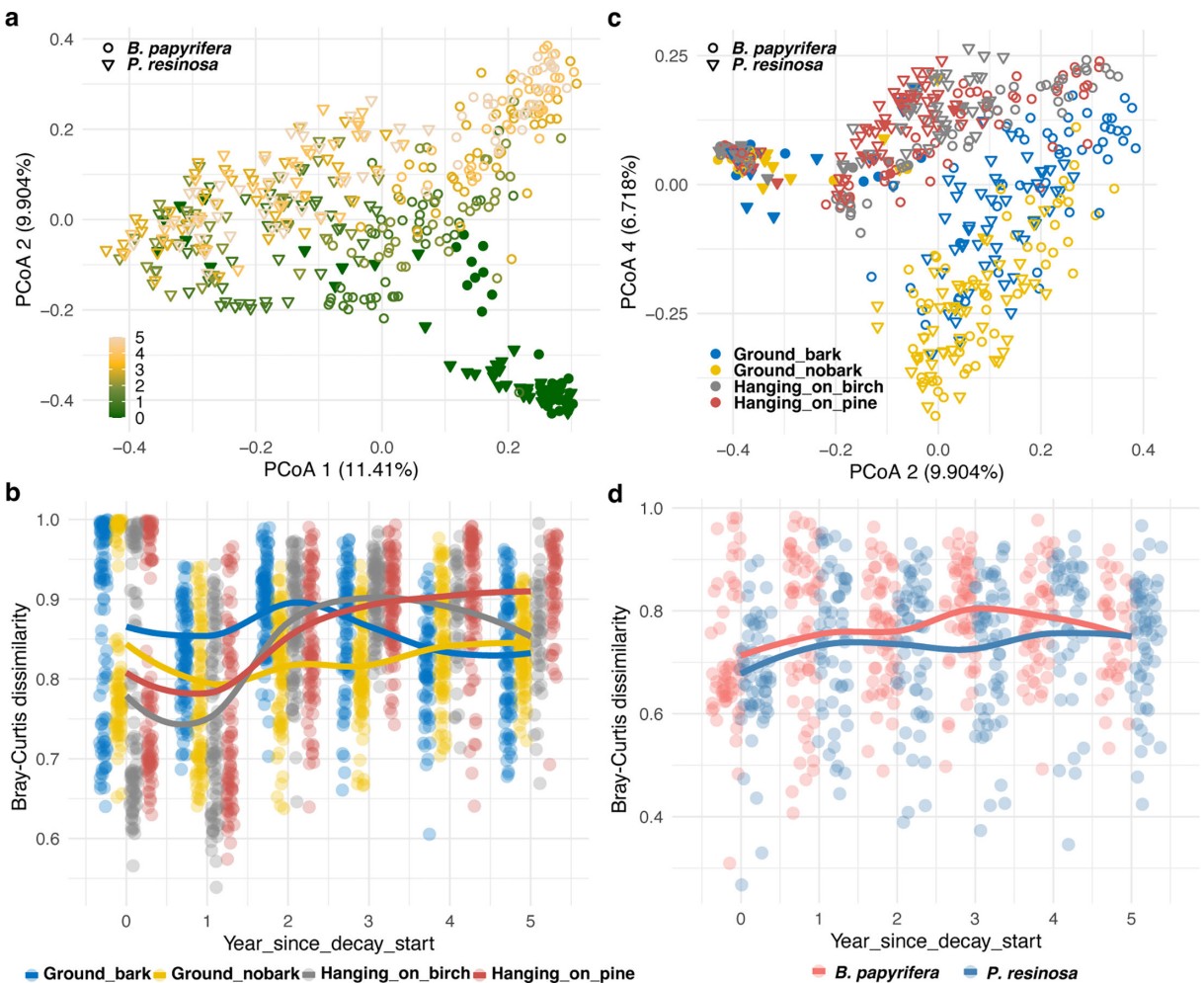

**FIG 3** Different fungal communities decaying *B. papyrifera* and *P. resinosa* in different treatments. (a) PCoA of Bray-Curtis dissimilarity of fungal communities between samples shaped by tree host and colored by time, whereas sound and decayed wood are represented by filled and open symbols, respectively. (b) Pairwise Bray-Curtis distances between *B. papyrifera* and *P. resinosa* within each decay time and each treatment. (c) PCoA of fungal community shaped by tree host and colored by treatments while using principal coordinate PCoA2 versus PCoA4. The sound and decayed wood samples are represented by filled and open symbols, respectively. (d) Pairwise Bray-Curtis distances between treatments within each decay time and each tree host.

treatment-dependent dominance was apparent (Fig. 4b). The treatment-dependent dominance can also be seen among the individual log replicates (Fig. S6 and S7).

We also assigned each OTU to a functional guild (Fig. S5b) and found white rot taxa dominating birch and pine decay communities, matching our L:D results showing wood white rot patterns. Brown and soft rot fungi were present in lower abundances in sound wood, but either did not proliferate or vanished over the course of succession, irrespective of treatment. In one of the birch trees (birch_10), the brown rot fungus *Fomitopsis betulina* (previously *Piptoporus betulinus*) preoccupied its territory in high amounts before decay started (Fig. S8a and b; average 23.6%, up to 55% in bark-off logs; absolute reads, 238,161). However, as decay progressed, it gradually lost its territory to other endophytic taxa, such as Agaricomycetes (SH185074.02FU) in aboveground logs and *Xenasmaella* in bark-off logs (Fig. S8b and c). Fungi listed in guilds other than wood rot types (white, brown, soft rot) included a large "saprotroph" presence, particularly at later decay stages of birch when Ascomycota, particularly in the order Helotiales (Fig. 4a), dominated. Pine supported a significant "plant pathogen" taxa presence, as well.

Given our repeated measures design, we could also statistically compare fungal abundances as a function of treatment and time (Fig. S9a and b). Bark-off birch was significantly higher than bark-on birch for genera *Xenasmatella*, *Plicaturopsis*,

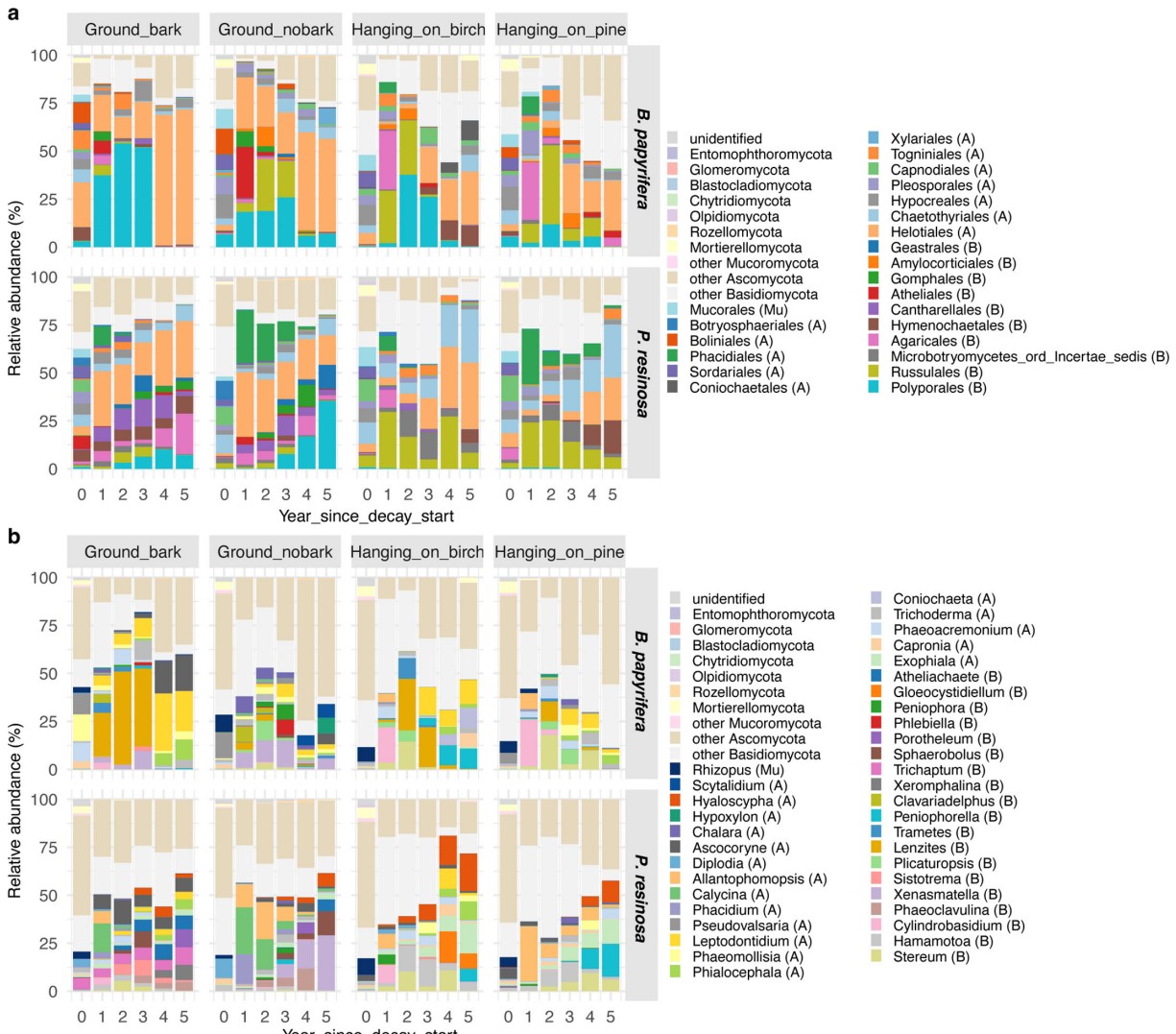

**FIG 4** Succession of fungal communities was treatment-specific in decaying both *B. papyrifera* and *P. resinosa*. Relative abundance of fungi (a) at the order level and (b) at the genus level along the decay process of *B. papyrifera* and *P. resinosa*. Only orders or genera with at least 7% relative abundance in one of the treatments were specified, and all others were classified to the phylum level. All the taxa that were not assigned to the level of order or genera were also classified to the phylum level. Abbreviations: A: Ascomycota; B: Basidiomycota; Mu: Mucoromycota.

*Xeromphalina*, *Trichaptum*, *Sphaerobolus*, *Phlebiella*, *Peniophora*, *Calycina*, *Chalara*, *Hypoxylon*, and *Scytalidium* but was significantly lower in *Sistotrema*, *Lenzites*, *Leptodontidium*, and *Ascocoryne*. Aboveground birch was significantly higher than bark-on birch in ground contact for genera *Stereum*, *Cylindrobasidium*, *Plicaturopsis*, *Phacidium*, and *Allantophomopsis* but was significantly lower in *Xenasmatella*, *Sistotrema*, *Clavariadelphus*, *Phlebiella*, *Peniophora*, *Ascocoryne*, and *Scytalidium*. Bark-off pine was significantly higher than bark-on pine for genera *Xenasmatella*, *Sphaerobolus*, *Peniophora*, *Calycina*, and *Allantophomopsis* but was significantly lower in *Sistotrema, Trichaptum, Phaeoacremonium*, *Phialocephala*, and *Leptodontidium*. Aboveground pine was significantly higher than bark-on pine in ground contact for genera *Stereum*, *Peniophorella*, and *Phaeomollisia*, but was significantly lower for genera *Sistotrema*, *Trichaptum*, *Sphaerobolus*, *Atheliachaete*, *Phacidium*, *Calycina*, and *Ascocoryne*. Broadly, the biological variability among endophytes from tree to tree for our 10 replicate trees disappeared (became more uniform) over time within treatments, but dominant fungi differed as a function of

treatment. For example, compared with the bark-on samples of birch or pine, *Xenasmatella* dominated in bark-off treatments while *Stereum* dominated in aboveground treatments.

## Endophyte persistence and treatment effects

One of our core hypotheses was that by tracking community development from time 0 in a repeated measures design, we would test whether endophytes could persist as the rule, not the exception, to become dominant OTUs in saprotroph communities. We also hypothesized that endophytes would exert more influence as emergent saprotrophs in treatments when limiting external colonizer accessibility (bark on >bark off; aboveground >ground contact). We tracked "endophyte influence" in two ways: first, by comparing among decaying wood treatments the read count percentage of OTUs that were initially present at time 0, and second, by linking dominant OTUs in decaying wood to their presence/absence at time 0. The first approach focused on the origins of all OTUs; the second approach focused on the origin of single OTUs in the top 10 fungi, collectively.

We found that fungal endophyte persistence decreased gradually over time in highly decayed birch, but not in pine, which was in earlier decay stages by year 5 (Fig. 5a). The read count percentage of endophyte fungal taxa averaged 74.2% for birch at year 2 and 76.8% for pine at year 5. The removal of bark decreased endophyte persistence (53.1%) when compared with ground contact bark-on samples (79.1%) in birch, not in pine (Fig. 5b). Hanging samples showed an unexpectedly lower endophyte persistence in birch, which was caused by *Lenzites* that were not detected in aboveground sound wood samples next to birch. However, *Lenzites* was a persistent endophyte in birch samples in all the other treatments (Fig. S10). This might be the caveat of low DNA abundance and read counts for endophytes in sound wood, where endophyte biomass per wood volume is very low. This might also be the caveat of the heterologous distribution of an endophyte within a tree. Focusing on the single, dominant OTUs, we found that most of the top 10 abundant fungi in decayed wood were OTUs that had been present as endophytes at very low read counts in sound wood (Fig. 5c; Fig. S10).

Although endophytes remained highly persistent in all treatments, the fungi rising to dominance in each treatment were treatment-specific once decay was initiated. The treatment effect overwrote endophyte heterogeneous distributions among tree replicates, discernible as Bray-Curtis distances in sound wood (between replicates) (Fig. S11).

## Bacterial community diversity

Our bacterial data set contained 18,931 OTUs, with 9,298 OTUs in sound wood and 17,159 in decayed wood, with an overlap of 7,526 OTUs. Similar to fungal patterns, bacterial species richness significantly increased in decayed wood compared with sound wood (Fig. S12a). Unlike fungi, however, the Shannon index did not decrease over decay time but instead increased (Fig. S12b). Ground contact logs had significantly higher bacterial richness and Shannon diversity than aboveground logs in birch or pine (Fig. S12c and d), and in birch, the bark-off logs had significantly higher bacterial diversity than bark-on samples.

Bacterial communities also did not diverge among treatments like fungal communities. Instead, bacterial communities that differed between sound birch and sound pine (Fig. S13a; $P = 0.001$, $R^2 = 0.079$) became similar as decay progressed (Fig. S13b; $P = 0.001$, $R^2 = 0.046$). The variance partitioned to the effect of decay time on bacterial communities was higher in birch than in pine (birch: $P = 0.001$, $R^2 = 0.092$; pine: $P = 0.001$, $R^2 = 0.066$), and treatment effect on bacterial community variation was relatively small ($P = 0.001$, $R^2 = 0.038$) (Table S3). Although this treatment effect was only shown in decayed wood (sound: $P = 0.192$, $R^2 = 0.037$; decayed: $P = 0.001$, $R^2 = 0.062$), the bacterial communities among treatments became similar as decay progressed (Fig. S13c and d).

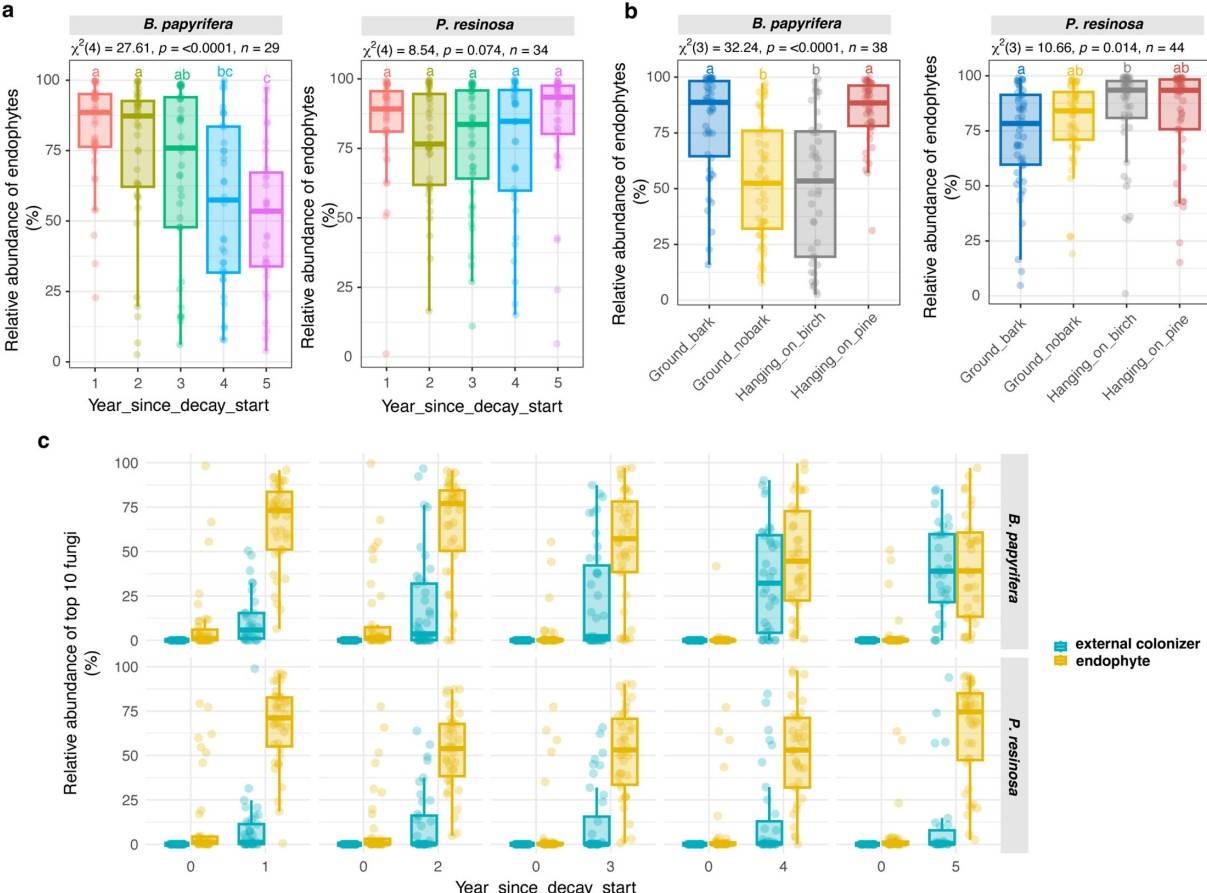

**FIG 5** Persistence of fungal endophytes to become the dominant fungi. The read count percentage of endophytes was compared as a function of (a) decaying time and as a function of (b) treatment. The *P* values indicate the significance test using the Friedman test, and the lowercase letters represent the significant difference among decaying time or treatment by the pairwise paired Wilcoxon test in *B. papyrifera* and *P. resinosa*, separately. (c) Read count percentage of endophytes and external colonizers in the top 10 abundant fungi within each decaying time and treatment. Their corresponding read count percentage at time 0 was shown.

At the phylum level for bacteria, we observed similarities in abundance patterns across different treatments and tree species (Fig. S14a). The deadwood bacterial communities were mainly dominated by Gammaproteobacteria, Alphaproteobacteria, Bacteroidetes, Actinobacteria, and Acidobacteria. Notably, at the phylum level, Gammaproteobacteria proliferated in birch but contracted in pine. Acidobacteria and Actinobacteria proliferated in both birch and pine. Alphaproteobacteria were abundant, but only shifted slightly (Fig. S14b). At order and genus levels (Fig. S15), a clear shift was apparent once decay was initiated, toward a significant presence of *Sphingomonas*, *Luteibacter*, *Pseudomonas*, and *Burkholderia*. Although more challenging to track dominant OTUs in this case, the endophyte read count persistence indicated that 67.6% of OTUs in decaying birch at year 2 and 75.1% of OTUs in decaying pine at year 5 were present as OTUs in sound wood.

## Co-occurrence of fungi and bacteria

Although community data showed patterns of divergent fungal communities with decreased diversity for fungi, seemingly decoupled from bacterial dynamics of convergence and stable/increased diversity, we conducted a co-occurrence network analysis to link occurrences on a log-by-log basis, again benefiting from a repeated measures design (Fig. S16). At the taxa level, we found that birch had fewer correlations than pine (52 and

113 nodes, respectively) and a similar average degree of connectivity (5.143 and 3.000, respectively) (Fig. S16a). Notably, no direct bacteria-fungus relationships were present in either birch or pine. Considering 861 fungal taxa and 2,672 bacterial taxa examined, this was surprisingly low. There were also no direct bacteria-fungus relationships at the genus level (Fig. S16b).

## DISCUSSION

Among the fungi we tracked over 5 decay years, Basidiomycota dominated (Fig. S5a), we assume, due to their ability to efficiently decompose recalcitrant wood biopolymers (63). Unlike birch, in which Polyporales were mainly responsible for decomposition, more diverse Basidiomycota, including Agaricales, Cantharellales, Hymenochaetales, and Gomphales, were associated with pine decomposition. In previous sporocarp (64, 65) and molecular surveys (66, 67), it was reported that wood-inhabiting fungi vary depending on the tree host, particularly between deciduous and coniferous wood, even if they are comparatively less host-specific than some other litter-degraders (18). The tree species preferences of fungal decomposers are more robust in broadleaf trees (67).

Nesting genus within order graphs revealed patterns of fungal dominance that were, in some cases, not surprising, given observed fruiting at the time of sampling. For example, the dominance of *Lenzites betulina* (synonym *Trametes betulina*) in bark-on birch logs in ground contact over the first 3 years (Fig. 4b) matches the order Polyporales dominance over the same years (Fig. 4A), and we observed abundant *L. betulina* sporophores in the field in this treatment. Similarly, *Stereum hirsutum* (order Russulales) was an abundant fruiter on aboveground logs for both tree species. We could even make connections in individual log replicates (Fig. S6 and S7); for example, the abundant *Trichaptum abietinum* (order Hymenochaetales) sporophore production observed in the field on one replicate pine log (pine_07; ground contact bark-on) matched genus- and order-level data. However, most taxa were not observed fruiting and would not be "visible" in culture-dependent analyses, making this repeated measures data set extremely powerful for tracking fungi (and bacteria) across an arc of decay in relevant field situations as treatments designed in this study.

Although brown rot fungal sporocarps are common and visible on birch (e.g., *Fomitopsis betulina*) and pine (e.g., *Fomitopsis mounceae*) at this site, white rot was clearly the dominant process (L:D > 0.8) (Fig. 2c) and was the dominant rot-type guild (Fig. S5b) in this study. Only a few samples, especially from the aboveground treatment group, were identified as brown rot (Fig. 1c; L:D < 0.8). However, their fungal rot guild was found to mostly contain unknown saprotrophs and some white rot, which we assume might be simultaneous white rot fungi that remove lignin and carbohydrates at a similar rate. Brown rot fungi break down the wood structure in a random manner through a reactive oxygen species (ROS) reaction (68). This process releases more soluble sugars into the surrounding environment, making these sugars more accessible to other "cheaters." As a result, brown rot fungi are considered less competitive compared with the gradual depolymerization carried out by white rot fungi, which may explain the prevalence of white rot in our study. Notably, white rot fungus *Lenzites betulina* was the most abundant fungus in decaying ground-contact birch in this study (Fig. 4b), drawing a contrast with another white rot fungus, *Trametes versicolor,* dominance on ground-contact birch at the same research plot in a 2018 study using birch of similar dimensions (31). *L. betulina* has been identified as a combative secondary colonizer that is a mycoparasite of *T. versicolor* *(34)*. In this study, both *L. betulina* and *T. versicolor* were present as endophytes in sound birch, and *Trametes* has previously been noted for its endophytic strategies (31); hence, it is unclear whether *L. betulina* succeeded due to endophytic or mycoparasitic strategies.

*F. betulina* is a very common brown rot fungus on standing birch, latently present as an endophyte (69). Although the birch endophyte community has previously been shown in a lab trial to contain other viable brown rot endophytes in wood sampled at this forest site (30), *F. betulina* is typically the only brown rot outcome observed in standing birch in the field. However, the chances for *F. betulina* to win are typically low

after decayed wood advances to high decay stages or falls to the forest floor, compared with the competitive white-rot fungus (31), which is further supported by our study. *F. betulina* was present as an endophyte in sound wood (average absolute reads count, 197), and it even preoccupied territory in one birch tree (Fig. S8; birch_10; absolute reads, 238161), but it was all outcompeted by other white rot fungi, depending on the treatment group. Our study supported the idea that *F. betulina* is a stress-tolerant fungus and a primary colonizer during the early decay process, which is often replaced by competitive, secondary colonizers (34). Endophyte success varied among our treatments; we think to some degree by microclimatic variability created by treatment conditions (34, 35, 70, 71) as well as by inoculum potential of the endophytes (e.g., biomass).

As read count percentages, fungal endophyte persistence was seemingly high in all treatments, with dominant fungal taxa typically detected in low amounts in sound wood (Fig. 5). This bears the caveat, as mentioned previously for *Lenzites*, that an OTU present in decayed samples cannot, with certainty, be linked to the same inoculum present at time 0. Dominant fungal DNA may come from later arriving individuals, but their presence at the outset of decay in so many cases suggests a common transition of life strategies from endophytic to saprotrophic in the same individual fungi. This would seem to be an advantageous competitive strategy to occupy territory and limit colonization by external saprotrophs (30, 31).

Unlike the fungi, bacterial communities in our field study were shaped primarily by time rather than tree host or treatment (Fig. S13). Deadwood bacterial communities have been previously reported to show tree species preference in Angiosperm and Gymnosperm (conifer) substrates (e.g., *Fagus sylvatica* and *Picea abies* [72]) and are often dominated by *Proteobacteria*, *Acidobacteria*, and *Actinobacteria* (39, 73). Although previous studies have indicated the association of bacteria with wood-decaying fungi and the interaction between bacteria and fungi in decaying deadwood, including competition and co-operation (39, 49), our data did not show a substantial selection of dominant fungi on specific bacteria, and the co-occurrence of specific fungi-bacteria was low (Fig. S16). The universal homogenization of bacterial community structure in the decayed wood was not as we hypothesized, at least between aboveground and ground contact samples (74). However, there is a possibility that the rot type of fungi might be correlated with certain bacteria, as the nutritional modes are completely different between white and brown rot fungi (40). In our study, it is also possible that the prevalence of white rot could have liberated sugars and other metabolites in similar ways (43, 44) and thus unified the outcome among bacteria. This suggests decoupled community assembly dynamics at the taxonomical level, and it supports the idea that fungal specialists support bacterial generalists in decomposing forest deadwood.

## Conclusion

Using a repeated measures design enabled us to track the succession of wood decomposer fungi and their associated bacterial communities over 5 years in a field trial. The gain and loss of individual taxa could thus be traced to previous years, enabling a measure of persistence that, in most cases, for the white rot fungi that dominated decay, could be traced back to their presence as an endophyte in sound wood. This suggests that the priority gained by the endophytic strategy of colonizing fungal saprotrophs may explain much of the variability of the "downstream" decay process that currently is hard to predict in forest carbon models. With bacterial communities seemingly decoupled from this process, endophytic fungi may offer an additional "plant" trait to assist typical litter traits (e.g., nitrogen content) in predicting wood decay rates/fates in nature.

## ACKNOWLEDGMENTS

The authors acknowledge the China National GeneBank for the support of sequencing library preparation and amplicon sequencing. Research was made possible by the support of the Conservation and the Environment grants program at The Andrew W.

Mellon Foundation (New York, NY). A doctoral dissertation fellowship of the University of Minnesota supported Song in some stages of research, along with the Minnesota Agricultural Experiment Station funding #MIN-12-087 for Schilling. Zhang was generously funded by the College of Biological Sciences at the University of Minnesota.

Y.Z., Z.P., Z.S., and J.S. conceived the study; Z.S. and J.S. coordinated and carried out fieldwork and samplings; Y.Z. processed the samples, performed physiochemistry analysis, and performed genomic DNA extraction; Z.P. and Z.S. performed the sequencing work; Y.Z. analyzed data and interpreted the results; Y.Z. drafted the manuscript; Y.Z., Z.P., Z.S., and J.S. revised the manuscript. All authors approved the final version. All authors have agreed to be responsible for all aspects of the work to ensure that inquiries regarding accuracy or integrity are appropriately investigated and resolved.

## AUTHOR AFFILIATIONS

[1]Department of Plant and Microbial Biology, University of Minnesota, Saint Paul, Minnesota, USA
[2]Section of Food Microbiology, Gut Health, and Fermentation, Department of Food Science, University of Copenhagen, Frederiksberg, Denmark
[3]BGI Research, Sanya, China
[4]Shenzhen Key Laboratory of Environmental Microbial Genomics and Application, BGI Research, Shenzhen, China

## AUTHOR ORCIDs

Yanmei Zhang  http://orcid.org/0009-0000-5210-9030
Jonathan S. Schilling  http://orcid.org/0000-0003-0810-3007

## FUNDING

| Funder | Grant(s) | Author(s) |
|---|---|---|
| Andrew W. Mellon Foundation | | Jonathan S Schilling |
| Minnesota Agricultural Experiment Station | MIN-12-087 | Jonathan S Schilling |
| College of Biological Science at the University of Minnesota | | Yanmei Zhang |
| Doctoral Dissertation Fellowship at University of Minnesota | | Zewei Song |

## AUTHOR CONTRIBUTIONS

Yanmei Zhang, Conceptualization, Data curation, Formal analysis, Investigation, Methodology, Resources, Software, Validation, Visualization, Writing – original draft, Writing – review and editing | Zhuobing Peng, Conceptualization, Data curation, Methodology, Resources, Writing – review and editing | Zewei Song, Conceptualization, Data curation, Methodology, Resources, Writing – review and editing | Jonathan S. Schilling, Conceptualization, Funding acquisition, Methodology, Project administration, Resources, Supervision, Validation, Writing – review and editing

## DATA AVAILABILITY

The raw sequencing data are archived in CNSA under accession ID CNP0000490. All the scripts for sequencing data analysis and generating the pictures with statistical analysis are openly available in the GitHub repository Zhang2025Cloquet.

## ADDITIONAL FILES

The following material is available online.

## Supplemental Material

**Supplemental Figures (mSystems00382-25-S0001.pdf).** Fig. S1 to S16.
**Supplemental Tables (mSystems00382-25-S0002.xlsx).** Tables S1 to S3.

## Open Peer Review

**PEER REVIEW HISTORY (review-history.pdf).** An accounting of the reviewer comments and feedback.

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
