## [Reviewer comments · mSystems]

Repeated measures of decaying wood reveal the success and influence of fungal wood endophytes

Yanmei Zhang, Zewei Song, Zhuobing Peng, and Jonathan Schilling

Corresponding Author(s): Jonathan Schilling, University of Minnesota

Review Timeline:

Submission Date:	March 16, 2025
Editorial Decision:	May 5, 2025
Revision Received:	June 20, 2025
Accepted:	July 10, 2025

Editor: Nhu Nguyen

Reviewer(s): Disclosure of reviewer identity is with reference to reviewer comments included in decision letter(s). The following individuals involved in review of your submission have agreed to reveal their identity: Yuanyuan Bao (Reviewer #2)

Transaction Report:

DOI: <https://doi.org/10.1128/msystems.00382-25>

Re: mSystems00382-25 (**Repeated measures of decaying wood reveal the success and influence of fungal wood endophytes**)

Dear Dr. Jonathan S Schilling:

Overall, the reviews to your manuscript was positive. All four reviewers had relatively minor comments/questions for you to address. A commonality among the reviewers was a need for clarification/additional details on the methods, and potential additional analyses.

I also suggest additional methods clarification for the co-occurrence network: Please include that you've analyzed the networks at both the OTU/"species" and Genus level. Note that "taxon/taxa" refers to any level of taxonomic classification and not just the species level as used in the manuscript; this should be corrected. State how you combined the fungi and bacteria data matrix, if and how you transform the data (a necessary requirement for doing multi-gene networks), whether relative or absolute abundance was used and in which context. If data transformation was not performed, please make sure to reanalyze and report updates on the network links.

It is important that you return the manuscript within 60 days; if you cannot complete the modification within this time period, please contact me. Additional time may require that the manuscript be rejected and resubmit for additional reviews. If you do not wish to modify the manuscript and prefer to submit it to another journal, notify me immediately so that the manuscript may be formally withdrawn from consideration by mSystems.

Revision Guidelines

Sincerely,
Nhu Nguyen

Reviewer #1 (Comments for the Author):

Comments

This paper reports the results of a five-year monitoring study for wood decay process and fungal-bacterial communities. The working efforts are massive, and the data are substantial. The results include some interesting points. However, I found some drawbacks in statistical comparison. Furthermore, typos, mismatches in figure legends, and discussion-like descriptions in the Results section made me feel that the manuscript was rather premature. These points should be fixed before publication.

L147 and somewhere in the text

"Discs" maybe referred as "cookies" in Fig.S1.

L199-205

These treatments can not remove PCR bias. The data should not be analyzed as quantitative data, unless the samples do not include internal standard mock DNA. If you keep fungal OTU data quantitatively, please show rationality.

L214

The effect of treatment on the carbohydrate loss?

ANOVA and Bonferroni-corrected pairwise t-test. Did you check normality and equal variances of the data? If the data do not have normal distribution and equal variances, non-parametric post hoc tests such as Steel-Dwass test are better to be applied. The significance appearance shown in Fig. 1b and Fig. S2 are dubious. Better to check the results compared by Steel-Dwass test instead of pairwise-paired Wilcoxon test.

L230, 231 Genus

Why capital letter?

L254 decay stage

This term is frequently referred in the text, but not clearly defined in the paper. Please define the threshold between the stages. Also, assigning samples which show L:D below 0.8 as brown rot might be discussion. Showing the L:D results of 2012 for birch samples to compare the results with that of pine in the same decay stage, which is fine. But what about the birch samples in 2015? No change from samples in 2012? In addition, please add the information about the Year_since_decay_start to the two figures in Fig.1c.

Fig. S5a,c

What is the difference between open and close symbols?

Caption's a,b,c,d are not correctly reflecting figure's a,b,c,d.

L293-308

The statements shown here includes Discussion.

Fig. S10

Please add the descriptions for abbreviations of ecological groups.

Fig. S17

Refer this figure in the Results section if you need it.

L407-409

What is the reason of this discrepancy?

Reviewer #2 (Comments for the Author):

This five-year study presents a rigorous and well-executed investigation into microbial succession in forest ecosystems. The long-term design enhances the reliability of findings, while the consistent sampling and data processing reflect significant technical effort. Its rarity and depth place it at the forefront of ecological research, offering valuable long-term insights. Importantly, it serves as a strong reference and inspiration for future long-term ecological studies. Although, in my point of view, this study is primarily descriptive, the value of the five years of in situ data, results, and observed patterns is nonetheless significant. It provides a solid research foundation for developing predictive models of carbon sequestered or released by wood

on Earth based on microbial functional traits.

Some suggestions for revisions are as follows:

In Figure 1, based on the x-axis, it appears as if the experiments for the two tree species were not initiated on the same day (although I assume they were). The x-axis looks like it is not a continuous variable. Bar plots, such as those in Fig. S4, seem to avoid this confusion. The authors might consider adjusting the presentation to avoid potential misunderstanding.

The resolution of most figures in the supplementary material is quite low, making it difficult to read the text and data clearly. I am not sure if this issue occurred during my download process. I suggest the authors verify the figure quality.

I really like Fig. S1, as it clearly illustrates the experimental setup of this study. I recommend the authors consider moving this figure into the main text. Of course, this is only a suggestion and entirely up to the authors.

Lines 177-175: The description of the DNA extraction method seems a bit vague. Was plant DNA also extracted during this process? Please clarify.

Line 184: The 17 decayed wood samples that failed amplification - which treatments and time points do they belong to? Some readers might be interested, so it would be helpful to specify.

In Fig. 1, there is a typo: "B. pyrarifera" should be corrected to "B. papyrifera."

Lines 246-253 and throughout the manuscript: Would it be better to consistently use either "birch" or "B. papyrifera" to make reading smoother? Currently, the text switches between the common and Latin names.

Fig. S16 and Lines 384-393: Since no network links were found between bacteria and fungi, could the lack of statistical correlation still reflect their real-world interactions? The authors should discuss this point. As mentioned in Lines 448-451, fungi may influence bacterial communities via metabolites, but the current network analysis suggests no direct statistical association, which needs explanation and discussion.

Lines 262-268: Are there dynamic changes in the physicochemical properties of wood over the five years of decay? Such data might be useful for readers.

Lines 394-398: The content here seems repetitive. It may not be ideal to place this repetition in the discussion section.

Lines 446-451: Considering the decoupled dynamics between fungal and bacterial communities and the authors' speculation on bacterial community convergence, it is recommended to supplement the analysis with potential environmental drivers or correlations between bacterial and fungal communities.

Reviewer #3 (Comments for the Author):

The paper described a five year into the decomposition of birch and pine logs that were grouped into four parts. One part was logs that were placed onto the ground and another part where the bark was removed which were also placed onto the ground. The other two parts were suspended in the air against the same tree species ie. Birch against birch, and against another tree species, ie. Birch against pine. Decomposition with birch occurred more quickly and the level of decomposition associated with birch at year two was similar to the level associated with pine at year five. Fungal diversity was high at the beginning but became reduced but since decomposition with pine was occurring more slowly, diversity generally remained high. Eventually, one species of fungus appeared to become dominant in birch.

The paper is interesting and well presented. My only major question is why microbial diversity associated with the sound wood was high. I would have expected this to be negligible or at least extremely low, perhaps due to introduction of microbes through cracks in the wood structure. Therefore, in section 155-162, there is a concern that 70% ethanol might not have removed some microorganisms resulting in cross contamination.

Line 143 What was the age of the trees?

L175 How much was used for DNA extraction?

Line 360 commas in the wrong place

References some have full journal titles and others are abbreviated.

Supplemental data Fuzzy and difficult to read.

Reviewer #4 (Comments for the Author):

Review for "Repeated measures of decaying wood reveal the success and influence of fungal wood endophytes"

This study aims to compare the temporal relationships between fungi and bacteria present in sound wood and those that are

external colonizers under multiple accessibility conditions. Data over 5 years, showed treatment specific trends and decoupled dynamics between bacteria and fungi. I believe their final statement that, "This suggests decoupled community assembly dynamics, and it supports the idea that fungal specialists support bacterial generalists in decomposing forest deadwood", is strongly supported by their data. Below are some comments/questions:

The authors state that 44% of sound wood samples had less than 1000 reads after chloroplast and mitochondrial read removal, and while transforming to relative abundance accounts for some differences in sequencing depth, it does not fully correct for differences in microbial load and may introduce/mask compositional biases when total microbial DNA is low. Did the authors find that the low microbial load samples have more 'noise' or were more inconsistent across samples? Low biomass could result in increased noise. Have the authors tried other normalization approaches or approaches such as repeatedly rarefying data as a way to capture all the data as described by Schloss (<https://journals.asm.org/doi/10.1128/msphere.00355-23>)?

Can the authors add a brief description of how the OTU table was generated? Can the authors describe the choice to use OTU picking methods over current ASV denoising methods?

Were the diversity metrics calculated from the OTU counts or relative abundances?

Line 269, couldn't richness difference be a by-product of your sequence depths? How can you be sure that some of the decayed wood OTUs were not rare, low abundance in the sound wood but not detected due to your low sequence output? For example, lower abundance taxa in the decayed wood could be in your sound wood but below the level of detection.

How evenly distributed were the overlapping OTUs across sound wood samples and decayed wood samples? And between your bioreps, were these similar or housed a general core community?

Response to Reviewers' comments (*Please find our responses highlighted in blue italics. The line numbers in our response correspond with those in the revised manuscript*):

Editor

I also suggest additional methods clarification for the co-occurrence network: Please include that you've analyzed the networks at both the OTU/"species" and Genus level. Note that "taxon/taxa" refers to any level of taxonomic classification and not just the species level as used in the manuscript; this should be corrected. State how you combined the fungi and bacteria data matrix, if and how you transform the data (a necessary requirement for doing multi-gene networks), whether relative or absolute abundance was used and in which context. If data transformation was not performed, please make sure to reanalyze and report updates on the network links.

We value your thoughtful feedback and suggestions regarding our study. As you recommended, we have clarified our methods for the co-occurrence network.

- 1) In the Methods section, we have added clarification in Lines 257-258: "The co-occurrence network analysis of fungi and bacteria was conducted at both the OTU and Genus levels, separately." Additionally, we have corrected the word "taxa" to "OTU" in the caption for Fig. S16.*
- 2) In the Methods section, we have added a statement to clarify our approach to data transformation, the merging of fungal and bacterial data matrices, and the construction of graphs for network visualization. Please check the Lines 258-266: "The OTU tables for fungi and bacteria were transformed to relative abundance separately. OTUs with low abundance, defined as appearing in fewer than five samples and having a relative abundance of less than 0.5%, were excluded for both fungi and bacteria. The resulting fungal and bacterial OTU tables were then merged on a per-sample basis. Next, pairwise Spearman correlations were calculated for the combined data, covering fungal-fungal, bacterial-bacterial, and fungal-bacterial associations. Only those correlations with estimates less than -0.8 or greater than 0.8, along with a Benjamini-Hochberg adjusted p-value of less than 0.01, were included in the network analysis. Finally, the correlation matrix served as the input for converting to an adjacency list using the 'igraph' R package. The same approach was applied at the Genus level."*

Reviewer #1 (Comments for the Author):

Comments

This paper reports the results of a five-year monitoring study for wood decay process and fungal-bacterial communities. The working efforts are massive, and the data are substantial. The results include some interesting points. However, I found some drawbacks in statistical comparison. Furthermore, typos, mismatches in figure legends, and discussion-like descriptions in the Results section made me feel that the manuscript was rather premature. These points should be fixed before publication.

We sincerely appreciate this positive feedback and the thoughtful comments regarding our study.

We have carefully considered each comment and made corresponding changes. Furthermore, the manuscript has undergone a thorough proofreading process to correct typographical errors and to ensure consistency in the figure legends.

L147 and somewhere in the text

"Discs" maybe referred as "cookies" in Fig.S1.

We have changed the term "cookies" to "discs" in Fig. S1 to ensure consistency with the terminology used in the main text.

L199-205

These treatments can not remove PCR bias. The data should not be analyzed as quantitative data, unless the samples do not include internal standard mock DNA. If you keep fungal OTU data quantitatively, please show rationality.

The reviewer's concerns are valid. Removing the non-fungal or non-bacterial reads (the nuclear, mitochondrial, and chloroplast DNA of the host tree) yielded fungal or bacterial reads in sound wood samples that were significantly lower than those in decayed wood. Most of the copies of ITS2 or 16S rRNA V4 in sound wood indeed belong to the host tree; more copies of ITS2 or 16S rRNA V4 can be recovered as fungi and bacteria in decayed wood because the host DNA (tree DNA) has degraded. Our previous qPCR experiments using fungal-specific primers NSII and 58A2R confirmed that sound wood contains fewer fungal DNA copies compared to decayed wood (Zhang et al., 2024 doi: [10.1002/mbo3.70007](https://doi.org/10.1002/mbo3.70007)). Thus, the observed differences in read counts from fresh wood to decayed wood are due to a real change in the balance between plant and microbial DNA copies, not from PCR biases. This is a common challenge within host-associated metabarcoding, where primers cannot discriminate between microbes and the host. This presents an even bigger challenge in shotgun sequencing, as the lack of any taxonomically discriminatory PCR step will make genome assembly more difficult.

To avoid the loss of valuable information from samples with higher read depths, we explored two normalization methods for the dataset: 1) transforming the data into relative abundance; 2) separately rarefaction samples with low reads in the control group where microbial DNA was low, alongside samples with higher reads where microbial DNA was high. For example, we performed rarefaction at a reading depth of 5193 for sound wood samples and a depth of 19,501 for decayed wood samples for the fungal sequencing data. The rarefaction was performed at a reading depth of 1056 for sound wood samples and a depth of 12,524 for decayed wood samples for the bacteria sequencing data. We implemented both methods to calculate alpha diversity indices, including richness and Shannon diversity. The trends observed in fungal and bacterial diversity throughout the decay process and the direction of treatment effects were similar. Ultimately, we opted for the relative abundance transformation in our results. Please check the new supplementary table, Table S1, for a comparison of fungal or bacterial alpha diversity using both methods. We have added a justification in Lines 217-222: "or 2) performed rarefaction with 100 iterations for sound wood samples (fungi: 5193; bacteria: 1056) and decayed wood samples (fungi: 19,501; bacteria: 12,524), separately. The trends observed in fungal and bacterial diversity throughout the decay process and the direction of treatment effects were similar when comparing these two data normalization methods (Table S1). We opted for the relative abundance transformation to avoid discarding additional information in the decayed samples with high reads."

L214

The effect of treatment on the carbohydrate loss?

ANOVA and Bonferroni-corrected pairwise t-test. Did you check normality and equal variances of the data? If the data do not have normal distribution and equal variances, non-parametric post hoc tests such as Steel-Dwass test are better to be applied. The significance appearance shown in Fig. 1b and Fig. S2 are dubious. Better to check the results compared by Steel-Dwass test instead of pairwise-paired Wilcoxon test.

Our samples were not independent due to the use of repeated measurements, which allowed us to monitor changes in each of the 10 tree replicates over time and across different treatments. This method helps assess bias that could arise from the tree replicates. For our analysis, we chose to use a one-way repeated ANOVA followed by a Bonferroni-corrected paired t-test, assuming the necessary conditions were met—these include checking for outliers, ensuring normality, and verifying sphericity. If any of these assumptions were not met, we then turned to a non-parametric method, employing the Friedman test followed by pairwise Wilcoxon tests.

For the carbohydrate loss data, we began by checking for the outliers, normality, and sphericity. After confirming that all assumptions were met, we performed a one-way repeated ANOVA. However, in response to the reviewer's suggestion, we have run the post-hoc tests using the Steel-Dwass test (the data are not paired across the 10 tree log replicates) as a comparison. We found that the significance levels were generally consistent with those obtained from the Bonferroni-corrected pairwise paired t-test. This additional analysis did not change our earlier conclusion in Lines 289-290: "but only birch had significant treatment effects on glucan loss, xylan loss, and lignin loss".

For density loss data, we performed the Friedman test followed by pairwise paired Wilcoxon tests, as the data are not normally distributed. Following the Reviewer's suggestion regarding Fig. 1b and Fig. S2, we have also run the post-hoc tests using the Steel-Dwass test (data are not paired across the 10 tree log replicates) as a comparison.

*In comparison to the pairwise Wilcoxon test presented in Fig. 1b, the results from the Steel-Dwass test (Figure R1) revealed similar treatment effects in *B. papyrifera*; however, these effects were no longer evident in *P. resinosa*. To strengthen our findings, we have decided to exclude the post-hoc tests for *P. resinosa*, as the Kruskal test returned a non-significant result ($p = 0.2$), as shown in Fig. 1b.*

*In comparison to the pairwise Wilcoxon test presented in Fig. S2, the results from the Steel-Dwass test (Figure R2) showed that the difference among treatment groups was similar in time to reach 10% and 25 % mass loss in *B. papyrifera*, but the significance was no longer observed in time to reach 50% mass loss in *B. papyrifera*. The significance test was the same in *P. resinosa*.*

In summary, to ensure that the samples were paired across the 10 tree log replicates and that treatment was the only independent variable, we have retained the original statistical analyses in our revision.

Figure R1. Treatment effects on density loss in *B. papyrifera* and *P. resinosa*. The *p* values are from the Friedman test for each tree host, and the lowercase letters are from post-hoc tests using the Steel-Dwass test.

Figure R2. Time to 10% mass loss, 25% mass loss, and 50% mass loss of *B. papyrifera* and *P. resinosa* for different treatments. The 'years remaining' residence times for each tree log were assessed by a Weibull residence. The *p* values indicate the significance test of treatment using the Friedman test. The significance is shown: * $p < 0.05$, ** $p < 0.01$, *** $p < 0.001$. The different lowercase letters indicate the significance of the Steel-Dwass test.

L230, 231 Genus

Why capital letter?

Thanks for correcting the typos. We have corrected them to “genus” in Lines 247 and 248.

L254 decay stage

This term is frequently referred in the text, but not clearly defined in the paper. Please define the threshold between the stages.

Also, assigning samples which show L:D below 0.8 as brown rot might be discussion.

Showing the L:D results of 2012 for birch samples to compare the results with that of pine in the same decay stage, which is fine. But what about the birch samples in 2015? No change from samples in 2012? In addition, please add the information about the Year_since_decay_start to the two figures in Fig.1c.

*The thresholds that define decay stages vary between different tree species. We calculated the density loss thresholds between decay stages for *B. papyrifera* and *P. resinosa*, based on the Bole decay classes 1 to 5 as outlined by Harmon et al.*

*https://www.nrs.fs.usda.gov/pubs/gtr/gtr_nrs29.pdf, which is detailed in Lines 170-171. To be precise, the specific thresholds of density loss (%) for defining these decay stages have been incorporated into the caption of Fig. 1c. “For *B. papyrifera*, decay class I (1.55-9.17%), decay class II (9.18-21.35%), decay class III (21.36-45.63%), decay class IV (45.64-70.83%), and decay class V (\geq 70.84%). For *P. resinosa*, decay class I (18.54-19.02%), decay class II (19.03-27.56%), decay class III (27.57-48.78%), decay class IV (40.79-68.29%), and decay class V (\geq 68.30%).”*

Samples with L:D below 0.8 in Fig. 1c are mostly from the aboveground logs. After reviewing the fungal ecological groups, we found that these samples contained unknown saprotrophs and white rot (probably simultaneous white rot, due to a low L:D). We have added the statements in the Discussion section, Lines 433-437: “. Only a few samples, especially from the aboveground treatment group, were identified as brown rot (Fig. 1c; $L:D < 0.8$). However, their fungal rot guild was found to mostly contain unknown saprotrophs and some white rot, which we assume might be simultaneous white rot fungi that remove lignin and carbohydrates at a similar rate.”

We also analyzed the birch samples in 2015 (Figure R3). We found that these samples had much lower L:D ratios compared to those birch samples in 2012, due to significant density loss. However, the L:D ratio of birch samples in 2015 were not reliable for distinguishing between the rot type. This is largely because over half of the samples (25/40) has reached decay class IV or beyond, which is outside of the typical range of decay class II/III to distinguish rot type by L:D value. Therefore, we decided not to include the L:D ratio of birch samples in 2015 in our result.

We have explained Year_since_decay_start in Fig. 1. “The x-axis represents the time since decay start (unit: Year).”

Fig. S5a,c

What is the difference between open and close symbols?

Caption's a,b,c,d are not correctly reflecting figure's a,b,c,d.

The open and closed symbols represent the sound and decayed wood samples, respectively. We have added their explanation in the Fig. S5 caption. The order of the captions a, b, c, and d has also been corrected.

L293-308

The statements shown here includes Discussion.

This part of statements has been moved to Discussion in Lines 419-430.

Fig. S10

Please add the descriptions for abbreviations of ecological groups.

The abbreviations of ecological groups have been added in the caption of Fig. S10.

Abbreviations: Sa, Saprotroph; W, White rot; E, Ectomycorrhizal fungi; P, Plant pathogen; -, Unknown."

Fig. S17

Refer this figure in the Results section if you need it.

We have moved the Fig. S17 with its description from Discussion to Results section in Lines 325-329 and renamed it to Fig. S8: "In one of the birch trees (birch_10), the brown rot fungus Fomitopsis betulina (previously Piptoporus betulinus) preoccupied its territory in high amounts before decay started (Fig. S8a, b; average 23.6%, up to 55% in bark-off logs; absolute reads, 238,161). However, as decay progressed, it gradually lost its territory to other endophytic taxa,

such as Agaricomycetes (SH185074.02FU) in aboveground logs and Xenasmaella in bark-off logs (Fig. S8b, c)."

We have renamed the following supplementary Figures accordingly.

L407-409

What is the reason of this discrepancy?

We have added an explanation of white rot prevalence in our study in the Discussion section, in Lines 437-441: "Brown rot fungi break down the wood structure in a random manner through a reactive oxygen species (ROS) reaction. This process releases more soluble sugars into the surrounding environment, making these sugars more accessible to other "cheaters". As a result, brown rot fungi are considered less competitive compared to the gradual depolymerization carried out by white rot fungi, which may explain the prevalence of white rot in our study."

Reviewer #2 (Comments for the Author):

This five-year study presents a rigorous and well-executed investigation into microbial succession in forest ecosystems. The long-term design enhances the reliability of findings, while the consistent sampling and data processing reflect significant technical effort. Its rarity and depth place it at the forefront of ecological research, offering valuable long-term insights. Importantly, it serves as a strong reference and inspiration for future long-term ecological studies. Although, in my point of view, this study is primarily descriptive, the value of the five years of in situ data, results, and observed patterns is nonetheless significant. It provides a solid research foundation for developing predictive models of carbon sequestered or released by wood on Earth based on microbial functional traits.

We appreciate this positive feedback and thoughtful comments on our study. We have carefully considered each suggestion and made the necessary changes.

Some suggestions for revisions are as follows:

In Figure 1, based on the x-axis, it appears as if the experiments for the two tree species were not initiated on the same day (although I assume they were). The x-axis looks like it is not a continuous variable. Bar plots, such as those in Fig. S4, seem to avoid this confusion. The authors might consider adjusting the presentation to avoid potential misunderstanding.

We have changed Fig. 1a to be a bar plot per the reviewer's suggestion. Additionally, we have included the Wilcoxon test to compare the density loss difference between birch and pine at each sampling time. The caption has been modified accordingly to reflect these changes.

The resolution of most figures in the supplementary material is quite low, making it difficult to read the text and data clearly. I am not sure if this issue occurred during my download process. I suggest the authors verify the figure quality.

We have examined the quality of the supplementary Figures to ensure they meet the Journal's standards.

I really like Fig. S1, as it clearly illustrates the experimental setup of this study. I recommend the authors consider moving this figure into the main text. Of course, this is only a suggestion and entirely up to the authors.

Thank you for the recommendation. We've moved Fig. S1 to Fig. 1 in the main text to help readers better grasp our experimental designs and setup before diving into the main narrative. Consequently, we have rearranged the order of the Figures and supplementary Figures accordingly.

Lines 177-175: The description of the DNA extraction method seems a bit vague. Was plant DNA also extracted during this process? Please clarify.

We chose not to provide a detailed description of the DNA extraction methods, as we recently published a dedicated paper on this topic, which we reference in this manuscript (Line 180). The host DNA was extracted concurrently with microbial DNA during the process. Our findings indicated that sound wood samples contained a substantial amount of host DNA, including nuclear, mitochondrial, and chloroplast DNA. In contrast, decayed wood samples had a reduced level of host DNA, likely due to degradation.

Line 184: The 17 decayed wood samples that failed amplification - which treatments and time points do they belong to? Some readers might be interested, so it would be helpful to specify.

We have checked the 17 samples and found their distribution among treatment and time points, as below. Among them, one sample was lost during the sample storage process.

In B. papyrifera, ground_bark treatment: 1 sample in 2013, 2 samples in 2014

In B. papyrifera, hanging_on_birch treatment: 2 samples in 2014, 4 samples in 2015

In B. papyrifera, hanging_on_pine treatment: 2 samples in 2015

In P. resinosa, hanging_on_birch treatment: 4 samples in 2014

In P. resinosa, hanging_on_pine treatment: 1 sample in 2014; 1 sample in 2011 (Lost)

We found no pattern among treatments and time points for those failed samples. We have added a statement in Lines 195-196: "with no pattern among treatments or time points to explain amplification failures."

In Fig. 1, there is a typo: "B. pyparifera" should be corrected to "B. papyrifera."

Thank you. We have fixed the typo in the new Fig. 2.

Lines 246-253 and throughout the manuscript: Would it be better to consistently use either "birch" or "B. papyrifera" to make reading smoother? Currently, the text switches between the common and Latin names.

We have consistently used "B. papyrifera" in the Figures and their captions, but we have used "birch" throughout the manuscript to make the narrative clean. We have specified common and Latin names in the abstract (Line 32).

Fig. S16 and Lines 384-393: Since no network links were found between bacteria and fungi,

could the lack of statistical correlation still reflect their real-world interactions? The authors should discuss this point. As mentioned in Lines 448-451, fungi may influence bacterial communities via metabolites, but the current network analysis suggests no direct statistical association, which needs explanation and discussion.

Yes, we agree that the lack of occurrence pattern of fungi and bacteria at the OTU or Genus level does not necessarily imply that they do not have interactions at other levels. We suspected that there might be a correlation between bacteria and fungi at the fungal rot type level (white, brown rot, soft rot), but we could not confirm our hypothesis, as white rot fungi were prevalent in our study. We have added our explanations in the Discussion in Lines 479-481: “However, there is a possibility that the rot type of fungi might be correlated with certain bacteria, as the nutritional modes are completely different between white and brown rot fungi.”

Lines 262-268: Are there dynamic changes in the physicochemical properties of wood over the five years of decay? Such data might be useful for readers.

We measured the density loss over five years of decay, then focused on the birch samples in 2012 and the pine samples in 2015 to compare L:D and carbohydrate loss. We have removed the statement “Tracked across 5 years of annually-collected samples”.

Lines 394-398: The content here seems repetitive. It may not be ideal to place this repetition in the discussion section.

We have deleted this repetitive part as suggested by the Reviewer.

Lines 446-451: Considering the decoupled dynamics between fungal and bacterial communities and the authors' speculation on bacterial community convergence, it is recommended to supplement the analysis with potential environmental drivers or correlations between bacterial and fungal communities.

We suspected rot type might be a driver for correlations between bacteria and fungi. We have performed a co-occurrence analysis between fungal rot type and bacterial genus, but no association was found. However, we only had limited brown rot fungi in our data, which makes this co-occurrence analysis at the fungal rot type level suspicious.

We agree with the reviewer that there might be other environmental drivers for fungal or bacterial communities. However, we didn't test the abiotic factors, such as nitrogen, pH, moisture, etc., which might play a role in shaping their communities as shown by other studies.

Reviewer #3 (Comments for the Author):

The paper described a five year into the decomposition of birch and pine logs that were grouped into four parts. One part was logs that were placed onto the ground and another part where the bark was removed which were also placed onto the ground. The other two parts were suspended in the air against the same tree species ie. Birch against birch, and against another tree species, ie. Birch against pine. Decomposition with birch occurred more quickly and the level of decomposition associated with birch at year two was similar to the level associated with pine at

year five. Fungal diversity was high at the beginning but became reduced but since decomposition with pine was occurring more slowly, diversity generally remained high. Eventually, one species of fungus appeared to become dominant in birch.

The paper is interesting and well presented. My only major question is why microbial diversity associated with the sound wood was high. I would have expected this to be negligible or at least extremely low, perhaps due to introduction of microbes through cracks in the wood structure. Therefore, in section 155-162, there is a concern that 70% ethanol might not have removed some microorganisms resulting in cross contamination.

We sincerely appreciate your positive feedback and all the thoughtful comments on our study. We have carefully considered each comment and made corresponding changes.

Some fungi and bacteria are latent as endophytes in the sound wood. Even though their total DNA might be low due to their low biomass, the species richness is often high (tens to hundreds of OTUs). This has also been confirmed in other studies.

We recently published a dedicated paper on DNA extraction from wood samples to study microbial communities, which we reference in this manuscript (Line 180). In this paper, we also describe a thorough method to process samples before DNA extraction to minimize DNA contamination. I have included a short statement in the Lines 161-165: “To reduce the risk of contamination, we started by thoroughly cleaning the workbench and tools used for cutting wood, such as handsaw blades and chisels, with DNA AWAY surface decontaminant (Thermo Scientific, Waltham, MA, USA), followed by 70% ethanol between uses. The scalpels and 1/8-inch diameter drill bits were sterilized at 180°C for 4 hours.”

We carefully removed bark from each disc using a sterile scalpel before drilling wood sawdust for DNA extraction to minimize contamination. Our results showed that birch and pine hosted distinctive fungal and bacterial communities in sound wood (Fig. 3, 4; Fig. S13, 15), which supports the notion that the risk of cross-contamination is low in our processing.

Line 143 What was the age of the trees?

Although counting annual rings is challenging in these species, the tree diameters of 7-9 cm at the base, over 2.5 m in height, would indicate they were over 10 years old. We have added this detail in Line 144: “(> 10 years old)”

L175 How much was used for DNA extraction?

We used 400-500 mg of wood sawdust to extract DNA. We have added it to the Methods section shown in Line 179: “from 400 to 500 mg of sawdust”.

Line 360 commas in the wrong place

Thank you. This has been corrected.

References some have full journal titles and others are abbreviated.

We have checked the reference style according to the Journal’s standards. The journal title has been adjusted to the abbreviated form.

Supplemental data Fuzzy and difficult to read.

We have reviewed the quality of the supplementary Figures to ensure they meet the Journal's standards.

Reviewer #4 (Comments for the Author):

Review for "Repeated measures of decaying wood reveal the success and influence of fungal wood endophytes"

This study aims to compare the temporal relationships between fungi and bacteria present in sound wood and those that are external colonizers under multiple accessibility conditions. Data over 5 years, showed treatment specific trends and decoupled dynamics between bacteria and fungi. I believe their final statement that, "This suggests decoupled community assembly dynamics, and it supports the idea that fungal specialists support bacterial generalists in decomposing forest deadwood", is strongly supported by their data.

We sincerely appreciate this positive feedback and all the thoughtful comments on our study. We have carefully considered each comment and made corresponding changes.

Below are some comments/questions:

The authors state that 44% of sound wood samples had less than 1000 reads after chloroplast and mitochondrial read removal, and while transforming to relative abundance accounts for some differences in sequencing depth, it does not fully correct for differences in microbial load and may introduce/mask compositional biases when total microbial DNA is low. Did the authors find that the low microbial load samples have more 'noise' or were more inconsistent across samples? Low biomass could result in increased noise. Have the authors tried other normalization approaches or approaches such as repeatedly rarefying data as a way to capture all the data as described by Schloss (<https://journals.asm.org/doi/10.1128/msphere.00355-23>)?

Yes, we found that the sound wood samples with low microbial DNA had high "noise" among tree replicates. As shown in Table S2, the effect of tree log replicates is higher in sound wood than decayed wood samples both for fungi (sound: $p = 0.001$, $R^2 = 0.383$; decayed: $p = 0.001$, $R^2 = 0.093$) and bacteria (sound: $p = 0.002$, $R^2 = 0.269$; decayed: $p = 0.001$, $R^2 = 0.067$).

We have applied the rarefaction process separately to sound wood samples at a reading depth of 5193 for fungal data and 1056 for bacterial data, whereas for decayed samples, the depths were 19,501 for fungi and 12,472 for bacteria. The rarefaction was performed 100 times to calculate the mean value of alpha diversity. Although the species richness was different between the normalization methods of relative abundance and separate rarefaction, the trends observed in fungal and bacterial diversity throughout the decay process and the direction of treatment effects were similar. Ultimately, we opted for the relative abundance transformation in our results. Please check the new supplementary table, Table S1, for a comparison of fungal or bacterial alpha diversity using both methods. We have added a justification in Lines 217-222: "or 2) performed rarefaction with 100 iterations for sound wood samples (fungi: 5193; bacteria: 1056) and decayed wood samples (fungi: 19,501; bacteria: 12,524), separately. The trends observed in fungal and bacterial diversity throughout the decay process and the direction of treatment effects were similar when comparing these two data normalization methods (Table S1). We opted for

the relative abundance transformation to avoid discarding additional information in the decayed samples with high reads.”

Can the authors add a brief description of how the OTU table was generated? Can the authors describe the choice to use OTU picking methods over current ASV denoising methods?

We have added a brief description in the Methods section in Lines 185-192: “Briefly, the sequencing adaptors and primers were first removed by cutadapt, and then a 41-bp tail representing the large subunit (LSU) was removed from the end of the reverse primer. Sequencing reads with low quality (maximum expected error rate >1 and minimum length < 100 bp) were discarded by Vsearch. The forward reads R1 and the reverse reads R2 were separately aligned to SILVA(v132) or UNITE (v7.2) database, with a 97% similarity in best mode using BURST. R1 and R2 alignment were compared to generate a consensus alignment, which was used to calculate the operational taxonomic unit (OTU) table with taxonomic assignment.”

The reason for not using ASV is: 1) Fungal ITS sometimes spans a long distance, so some amplicons cannot be pair-end merged; 2) In order to utilize both R1 and R2 reads, we align both reads to references, and generate a consensus alignment based on R1 and R2 alignment; 3) If using ASV approach, amplicons with longer ITS region will be abandoned before analysis.

Were the diversity metrics calculated from the OTU counts or relative abundances?

As we mentioned in the first question/response, we calculated the diversity metrics using the relative abundances.

Line 269, couldn't richness difference be a by-product of your sequence depths? How can you be sure that some of the decayed wood OTUs were not rare, low abundance in the sound wood but not detected due to your low sequence output? For example, lower abundance taxa in the decayed wood could be in your sound wood but below the level of detection.

We acknowledge that the low species richness in sound wood might result from the low sequencing depth. The sequencing depth difference between sound wood and decayed wood was a result of a different ratio of microbial DNA and host wood DNA, which can be considered a biological fact. As explained in the first question/response, we tried to use two methods to do the normalization: 1) transforming the data into relative abundance; 2) separately rarefaction samples with low reads in the control group where microbial DNA was low, alongside samples with higher reads where microbial DNA was high. However, both methods can't rule out the possibility that some OTUs of decayed wood samples might not be detectable in sound wood samples due to low reading depth.

Additionally, we applied the rarefaction process at the same depth for all samples (fungi: 5,193; bacteria: 1,056). We found that the trends observed in fungal and bacterial diversity throughout the decay process and the direction of treatment effects were similar to the other two methods. However, the richness was much lower due to the significant loss of rare OTUs, especially in the decayed wood sample.

Furthermore, we have found that the read count percentage of OTUs in decayed wood that were initially present in sound wood as endophytes is high (Fig. 5a), reaching 74.2% for birch at year

2 and 76.8% for pine at year 5 (Line 362). This number is already high without considering the above possibility.

How evenly distributed were the overlapping OTUs across sound wood samples and decayed wood samples? And between your bioreps, were these similar or housed a general core community?

The overlapping OTUs were not evenly distributed across sound wood samples and decayed wood samples. As shown in Fig. 5c, most of the top 10 abundant fungi in decayed wood were OTUs that had been present as endophytes at very low read counts in sound wood. Additionally, our analysis of overlapping OTUs has confirmed that they were not evenly distributed across sound wood samples and decayed wood samples, as determined by their Bray-Curtis distances (Figure R4). The PERMANOVA analysis further supported that decay status had a significant effect on the overlapping communities both for fungi (birch: $p = 0.001$, $R^2 = 0.091$; pine: $p = 0.001$, $R^2 = 0.072$) and bacteria (birch: $p = 0.001$, $R^2 = 0.083$; pine: $p = 0.001$, $R^2 = 0.074$).

The effect of tree log replicates is higher in sound wood than decayed wood samples both for fungi (sound: $p = 0.001$, $R^2 = 0.386$; decayed: $p = 0.001$, $R^2 = 0.094$) and bacteria (sound: $p = 0.001$, $R^2 = 0.269$; decayed: $p = 0.001$, $R^2 = 0.068$). Therefore, a general core community was housed among replicates, especially in sound wood samples.

Re: mSystems00382-25R1 (**Repeated measures of decaying wood reveal the success and influence of fungal wood endophytes**)

Dear Dr. Jonathan S Schilling:

Thank you for responding to the reviewers comments and addressing those suggestions in clear detail. I have carefully reviewed your point by point responses and have determined that you have adequately addressed the suggestions provided by the reviewers and myself. As the comments, suggestions, and questions from the reviewers were generally minor and do not include any substantial flaws that require attention, I have decided that another round of peer review is not required.

As such, your manuscript has been accepted, and I am forwarding it to the ASM production staff for publication. Your paper will first be checked to make sure all elements meet the technical requirements. ASM staff will contact you if anything needs to be revised before copyediting and production can begin. Otherwise, you will be notified when your proofs are ready to be viewed.

Sincerely,
Nhu Nguyen
Editor
mSystems